# YKL-40 as a Potential Biomarker for the Differential Diagnosis of Alzheimer’s Disease

**DOI:** 10.3390/medicina58010060

**Published:** 2021-12-30

**Authors:** Ioannis Mavroudis, Rumana Chowdhury, Foivos Petridis, Eleni Karantali, Symela Chatzikonstantinou, Ioana Miruna Balmus, Iuliana Simona Luca, Alin Ciobica, Dimitrios Kazis

**Affiliations:** 1Department of Neurology, Leeds Teaching Hospitals, NHS Trust, Leeds LS2 9JT, UK; i.mavroudis@nhs.net (I.M.); rumana.chowdhury@nhs.net (R.C.); 2Third Department of Neurology, Aristotle University of Thessaloniki, 541 24 Thessaloniki, Greece; f_petridis83@yahoo.gr (F.P.); lena.kar@outlook.com (E.K.); melina.chatzik@gmail.com (S.C.); dimitrios.kazis@gmail.com (D.K.); 3Department of Exact Sciences and Natural Sciences, Institute of Interdisciplinary Research, ”Alexandru Ioan Cuza” University of Iasi, Alexandru Lapsuneanu Street, No. 26, 700057 Iasi, Romania; ioana.balmus@uaic.ro; 4Department of Biology, Faculty of Biology, “Alexandru Ioan Cuza” University, Carol I Avenue, No. 20A, 700505 Iasi, Romania

**Keywords:** YKL-40, biomarker, Alzheimer’s disease

## Abstract

Alzheimer’s disease (AD) is a progressive neurodegenerative disorder, associated with extensive neuronal loss, dendritic and synaptic changes resulting in significant cognitive impairment. An increased number of studies have given rise to the neuroinflammatory hypothesis in AD. It is widely accepted that AD brains show chronic inflammation, probably triggered by the presence of insoluble amyloid beta deposits and neurofibrillary tangles (NFT) and is also related to the activation of neuronal death cascade. In the present study we aimed to investigate the role of YKL-40 levels in the cerebrospinal fluid (CSF) in the diagnosis of AD, and to discuss whether there are further potential roles of this protein in the management and treatment of AD. We conducted an online search on PubMed, Web of Science, and the Cochrane library databases from 1990 to 2021. The quantitative analysis showed that the levels of YKL-40 were significantly higher in Alzheimer’s disease compared to controls, to mild cognitive impairment (MCI) AD (MCI-AD) and to stable MCI. They were also increased in MCI-AD compared to stable MCI. The present study shows that the CSF levels of YKL-40 could be potentially used as a biomarker for the prognosis of mild cognitive impairment and the likelihood of progression to AD, as well as for the differential diagnosis between AD and MCI.

## 1. Introduction

Alzheimer’s disease is a progressive neurodegenerative disorder, associated with extensive neuronal loss, dendritic and synaptic changes resulting in significant cognitive impairment [1,2,3]. The number of Alzheimer’s disease cases is expected to reach 75 million in 2030 and over 130 million in 2050, and it is the most common cause of dementia in the Western world [4]. The neuropathological hallmark of AD is the deposition of intracellular neurofibrillary tangles (NFT) and extracellular amyloid plaques [2]. The former consists of different forms of phosphorylated Tau, and the latter of amyloid beta [2]. An increased number of studies have given rise to the neuroinflammatory hypothesis in AD. It is widely accepted that AD brains show chronic inflammation [5,6,7], probably triggered by the presence of NFTs and insoluble amyloid beta deposits and is also related to the activation of neuronal death cascade [6]. Astrocytes and microglia are the main mediators of inflammation in AD brains. The microglial cells gather during a chemotactic reaction within the senile plaques in AD brains [8]. YKL-40, also known as chitinase 3-like protein 1 or human cartilage glycoprotein 39 is a chitin-binding lectin and belongs to the glycosyl hydrolase family 18 [9]. The human gene responsible for encoding YKL-40 was discovered in 1997 and is located in chromosome 1q31-q32 [10,11]. The expression of YKL-40 is related to inflammatory response and is involved in remodeling of the extracellular matrix [12]. The YKL-40 is expressed by macrophages, chondrocytes, neutrophils and synovial fibroblasts [13]. Furthermore, abundant expression of YKL-40 is present in astrocytes in neuroinflammatory conditions [9], and transcription of this protein can be induced by cytokines released by macrophages, resulting in morphological changes and altered mobility of astrocytes [9]. The levels of YKL-40’s mRNA were found to increase in both mouse models and human brains with AD [13].

Previous reports regarding the possible use of YKL-40 CSF levels in the diagnosis of AD focused on the comparison between healthy individuals versus AD or other neurodegenerative diseases patients [14,15,16]. However, a recent study regarding the CSF levels of YKL-40 in prodromal AD [17] demonstrated that YKL-40 CSF levels show differences in AD versus MCI patients, implying that this inflammatory marker could predict the conversion of MCI in AD, especially in the presence of APOE ε4 allele. Thus, based on these aspects, we aim to analyse the YKL-40 CSF levels power to predict neurodegeneration (specific to MCI, AD, or MCI-AD), and to evaluate its potential for differential diagnosis and therapeutical response.

## 2. Materials and Methods

We conducted an online search on PubMed, Web of Science, the Cochrane library databases from January 1990 to October 2021. We used the following terms: (“YKL-40” or “CHI3L1”) AND (“Alzheimer’s” or “AD”) OR (“Mild Cognitive impairment” or “MCI”) OR (“Dementia”) OR (“Neurodegenerative disorders” or “Neurodegeneration”. We only searched for studies written in the English language. We constructed a PRISMA flowchart of the search, with screening and eligibility following the PRISMA guidelines.

The inclusion criteria required case-control studies, and that all patients should meet the up-to-date diagnostic criteria. Studies with partial or incomplete data were not included.

We removed duplicate studies, reviews, or studies on familial disorders. Also, in vitro studies were excluded from the analysis. Two different authors extracted the data (MC and FP), and two additional authors cross-checked them (IM and EK). The levels of YKL-40 were analyzed in pg/mL, and if the values were given in different units, they were modified.

## 3. Statistical Analysis

For the statistical analysis and the graphical representation of the data, we used R Studio v. 1.4.1717-3 [18], the standardized mean difference, and a 95% confidence interval. We converted the medians and interquartile range to means and standard deviation using the formula provided by Hozo et al. [19]. The heterogeneity was assessed with the help of Cochrane’s Q statistic and I^2^. We used fixed or random effects models based on heterogeneity and plotted the random effects model using Forest plots. We investigated the influence of each study on the pooled mean difference using the meta-library in R Studio, and the omitting-one study method.

## 4. Publication Bias

Publication bias was assessed using funnel plots, and Egger’s test.

## 5. Quality Evaluation

For quality evaluation of the studies, we used the Newcastle–Ottawa quality assessment Scale to assess the quality of each included study, which was performed by two authors independently (FP and SC), with a third author consulted in case of discrepancies (EK). Three major components were collected: (1) the selection (0–4 points); (2) the exposure (0–3 points); (3) the comparability (0–2 points). The higher the score, the better the quality in methodology. All the studies that were included in this meta-analysis had scored greater than or equal to seven, indicating good qualities.

## 6. Description of Studies and Qualitative Findings

Rosen et al. (2014) evaluated the association of inflammatory CSF biomarkers with Alzheimer’s disease [20]. They used 25 AD patients and 25 control individuals, and they studied the CSF levels of chitotriosidase, the YKL-40 and monocyte chemoattractant protein-1. They found increased CSF levels of chitotriosidase and YKL-40 in the AD group, concluding that the YKL-40 levels may be helpful for the evaluation of cerebral inflammatory activity in AD patients. Hue Zhang et al. (2018) investigated the levels of YKL-40 in cerebrospinal fluid in 121 participants from the Alzheimer’s Disease Neuroimaging Initiative database [21]. They divided the participants into cognitively normal, stable mild cognitive impairment, progressive MCI, and Alzheimer’s dementia and they found that YKL-40 levels can be a predictor of AD diagnosis and are associated with hippocampal atrophy. They concluded that this protein may have utility for discriminating cognitively normal subjects and AD patients. Antonell et al. (2014) measured CSF YKL-40 levels in a cohort of 95 subjects, consisting of 43 controls, 18 pre-AD and 22 prodromal AD patients [22]. They reported differences between prodromal AD patients and controls, and some correlations with total tau and phosphorylated tau levels in the predementia AD group. They concluded that CSF YKL-40 levels could represent an important parameter to detect early pathophysiological changes in connection with the neurodegenerative process. Llorens et al. (2017a) comparatively analyzed YKL-40 levels in the brain and CSF samples from AD patients, vascular dementia patients, dementia with Lewy bodies/Parkinson’s disease, and sporadic CJD, then compared them with the control individuals [23]. They reported significantly increased YKL-40 levels in AD and sporadic CJD patients, concluding that increased YKL-40 is a disease-specific marker of neuroinflammation and that it might have a potential for application in the evaluation of therapeutic intervention in dementias with a neuroinflammatory component. Gispert et al. (2016) investigated the levels of YKL-40 in the CSF in 110 subjects, consisting of 49 controls, 27 mild cognitive impairment patients due to AD and 15 AD patients [24]. They reported a significant difference of the levels of YKL-40 between the groups of the study and a significant interaction in the association between YKL-40 levels and gray-matter volume. They reported a statistically significant increase of YKL-40 CSF levels in APOE ε4 carriers. Mattsson et al. (2017), analysed CSF samples from AD patients, patients with other dementias and MCI and healthy controls, finding higher chitotriosidase in the Alzheimer’s disease group, when compared to controls and patients with stable MCI, but they did not find a statistically significant difference in the YKL-40 levels between groups [25]. Hampel et al. (2018), studied the importance of six combined CSF biomarkers for the diagnosis of AD, including AB1-42, total-tau, phosphorylated tau, NFL, neurogranin and YKL-40 [26]. They used 21 controls, 35 AD patients, nine patients with frontotemporal dementia, and they reported significantly higher YKL-40 levels in AD patients in comparison to controls and FTD patients. Janelidze et al. (2015), measured CSF neurogranin and YKL-40 levels in a cohort of 338 individuals including cognitively healthy controls and patients with sporadic MCI, MCI which later developed AD, PD, DLB, VaD, and FTD patients. Alzheimer’s disease patients demonstrated considerably higher CSF levels of YKL-40 compared with DLB, but not with VaD and FTD. YKL-40 levels were not significantly different between sporadic MCI and MCI-AD patients [27]. Kester et al. (2015), examined the utility of YKL-40 in the CSF in 37 cognitively normal individuals, 61 MCI and 65 AD patients from the memory clinic-based Amsterdam Dementia Cohort [28]. They found higher YKL-40 levels in MCI and AD patients compared to controls, and they also reported that YKL-40 levels predicted progression from MCI to AD, suggesting that this protein could have an increased relevance in discriminating cognitively normal individuals from MCI and AD patients, and that YKL-40 levels may be associated with disease progression. Olsson et al. (2012) investigated the CSF YKL-40 levels in 96 AD patients, 65 healthy controls and 170 patients with MCI [29]. They reported increased YKL-40 CSF levels in AD patients compared to controls and in MCI patients who developed VaD, but not in MCI-AD patients. They also described stable YKL-40 levels in AD patients with a good correlation between YKL-40 and the cortical damage marker total-tau.

## 7. Qualitative Results and Statistical Analysis

### 7.1. AD vs. Normal Controls

The heterogeneity of the data in the comparison between AD patients and controls was high (I^2^ = 88.2%, τ^2^ = 170.06) in the eleven studies used, and therefore a random effects model was used which showed statistically significant difference between the two groups (z = 12.17, *p* < 0.0001) (Figure 1A). Influence analysis confirmed the robustness of the result (Figure 1B), and Egger’s linear regression test did not show any evidence of publication bias (Figure 2A).

### 7.2. MCI vs. Normal Controls

Six studies were included in the comparison between MCI and controls. The meta-analysis exhibited high heterogeneity with I^2^ at 97.1% and τ^2^ at 1529.33. The random effect model showed a z-value of 1.30 with a *p*-value of 0.1950 (Figure 1C). The results were not statistically significant. Influence analysis revealed that these results were not robust because omitting each one of the studies had a strong influence on the overall result (Figure 1D). Publication bias testing showed no evidence of publication bias (Figure 2B).

### 7.3. MCI AD vs. Normal Controls

YKL-40 levels were not significantly different in MCI AD and control groups. Although the heterogeneity was low, with I^2^ = 0.0%, the fixed effects model showed a z-value of 1.65 and a *p*-value of 0.0997 (Figure 3A).

### 7.4. MCIAD vs. MCI

For the comparison between patients with MCI that progressed to AD and stable MCI 5 studies were used. Statistical analysis showed low to moderate heterogeneity (I^2^ = 44.2%). We used a fixed effects model which showed that YKL-40 levels in the MCIAD group were significantly increased compared to the stable MCI group (z = 4.31, *p* < 0.0001) (Figure 3B). Influence analysis confirmed the robustness of the result and publication bias analysis did not reveal any evidence of bias (Figure 2C and Figure 3C).

### 7.5. AD vs. MCI

The comparison between Alzheimer’s disease patients and mild cognitive impairment showed high heterogeneity with I^2^ = 97.4%, and τ^2^ = 1969.97, and a z-value of 2.02 and a *p*-value of 0.0429 on seven studies (Figure 4A). YKL-40 was therefore significantly increased in AD patients compared to MCI patients. Influence analysis showed that the result was robust as omitting each study and recalculating the mean difference did not have a significant effect on the overall result (Figure 4B). Egger’s test did not show any evidence of publication bias (Figure 2D).

### 7.6. AD vs. MCIAD

Five studies were included in the comparison between Alzheimer’s disease patients and patients suffering from MCI that progressed to AD. The heterogeneity was moderate with I^2^ = 64.2% and τ^2^ = 360.25 (Figure 3C). We used a fixed effects model which showed a z-value of 7.68 (*p* < 0.0001), hence YKL-40 levels in the CSF of AD patients were significantly increased in comparison to MCI patients progressing to AD. The publication bias test did not reveal any evidence of bias (Figure 2E), and influence analysis showed that the result is robust (Figure 4D).

## 8. Discussion

One of the neuropathological features of AD is the deposition of neuritic plaques which consist of Ab deposits and are surrounded by microglia. Microglial cells play a crucial role as the main elements of the immune response in the brain and express multiple pro-inflammatory cytokines at molecular and cellular levels. Previous studies described substantially increased expression of mRNA of chitinase-3 like 3, a mouse homologue of YKL-40 on AD mice models in comparison to age-matched controls [13]. Similarly, in samples from AD human brains the levels of mRNA for YKL-40 and TNF-a were also significantly increased compared to controls [13].

Craig-Schapiro et al. (2010) described higher YKL-40 CSF levels in AD patients compared to controls, and progressive supranucleal palsy, however the exact results are not available, and this study was not included in the quantitative analysis of the present study [30].

YKL-40 levels were also increased in MCIs compared to early AD patients and normal controls, and they showed an inverse correlation with phosphorylated tau values. Another study found a nonlinear relationship between gray matter volume and the levels of YKL-40 in inferior and lateral temporal areas extending to the supramarginal gyrus, insula, inferior frontal cortex, and cerebellum in MCI and AD [31]. YKL-40 levels were also correlated with decreased cortical thickness in 27 AD patients compared to 80 control individuals [32]. Sutphen et al. (2015) described increased YKL-40 CSF levels in preclinical AD during middle age along other neuronal injury biomarkers, such as total tau, *p*-tau, and VILIP-1 [31].

Regarding the controversies and the reasons for the differences among individual studies, we can add that, for example Mattsson et al. (2017) described increased levels of YKL-40 in the AD patients, when compared to controls and patients with stable MCI, but they did not find a statistically significant difference between groups (as mentioned above), while also different markers analyzed by this research group did not correlate with each other, except for YKL-40 versus CCL2 (chemokine C-C motif ligand 2) in patients with other dementias [25]. Also, there were no modifications of YKL-40 in AD or in prodromal AD, while in the Craig-Schapiro study from 2010 [30] the cohort was larger than in Mattsson study, with the reported group differences “being small with large overlaps, making the findings difficult to reproduce”, as stated by [25].

One of the neuropathological hallmarks of AD is neuritic plaques which consist of fibrous deposits of Ab fragments and are surrounded by microglia. Microglial cells play a crucial role as the main components of the immune response in the brain and express multiple pro-inflammatory cytokines at mRNA and protein levels.

Thus, YKL-4 can be a reliable biomarker for the diagnosis of AD and the differentiation from MCI and MCI-AD. Furthermore, the increased YKL-40 levels in the CSF of AD patients demonstrate the significant role of neuroinflammation and microglia in the pathophysiology of the disease. The CSF levels of YKL-40 were also previously demonstrated as being increased in the CSF of MCI-AD patients compared to stable MCI, therefore this protein could be used as a biomarker for the prognosis of mild cognitive impairment and the likelihood of progression to AD [21].

## 9. Therapeutic Strategies Targeting YKL-40 and Neuroinflammation

Accumulative evidence has highlighted the role of neuroinflammation in the pathophysiology of AD, giving rise to the hypothesis that novel anti-inflammatory therapies could potentially reverse the consequences of neurodegeneration. The use of non-steroidal anti-inflammatory drugs or glucocorticoids has been suggested by previous studies as potential beneficial therapeutic strategies in AD [33]. NSAIDs (non-steroidal anti-inflammatory drugs) block the conversion of prostaglandin H2 into thromboxane and other prostaglandins, and could potentially block inflammation at early stages of AD. Indomethacin was shown to be beneficial in previous studies on the results of psychometric tests and AD assessment scale [14,33]. Naproxen sodium, can also lead to reduced tau to Ab1-42 ratio in the CSF of AD patients [34], but studies of selective cyclooxygenase-2 inhibitors failed to show any positive effect on AD patients [35].

## 10. Limitations

The present study showed interesting results on the role of YKL-40 CSF levels in the diagnosis of AD and the prognosis of MCI, however there are certain limitations. We only used studies written in English, so studies written in other languages may have been missed. Furthermore, the available data for the comparisons between MCI and MCI-AD and MCI and control groups were limited, which makes the reliability of the results on these comparisons weaker. Further studies required to extract safe results on the role of YKL-40 in the diagnosis of AD.

## 11. Conclusions

The present study demonstrated the usefulness of YKL-40 CSF levels in the diagnosis of AD and the differentiation from MCI, and has also shown that YKL-40 levels can be used, along with other biomarkers for the prognosis of MCI and the likelihood of progression to AD. The important role of neuroinflammation in the pathogenesis of AD has also been confirmed. Novel therapeutic strategies targeting neuroinflammation could potentially be beneficial for AD patients and could block inflammation underlying AD at early stages.

## Figures and Tables

**Figure 1 medicina-58-00060-f001:**
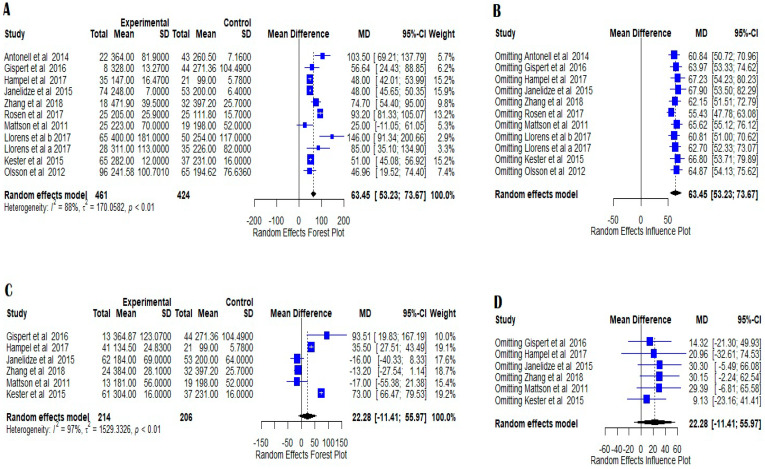
Forest plots of YKL-40 CSF levels in AD versus NC; (**A**) and MCI versus NC; (**C**) indicating high heterogeneity in random effect model and statistically significant differences between the compared groups. The figures indicate the means, standard errors (pg/mL), 95% confidence intervals, and weighted mean differences for each study included in this meta-analysis. For each comparison, a supplemental random effects influence plot was designed to further evaluate the possible publication bias when omitting each of the included studies (**B**,**D**), mean differences and 95% confidence intervals being calculated in each situation. No statistically significant impact was observed when omitting each of the evaluated studies. The standardized mean differences between patients and controls are represented as squares whose sizes are proportional to the sample sizes. The whiskers represent the 95% confidence intervals. The diamonds represent the pooled estimate based on the random-effect model with the center representing the point estimate and the width indicating the assessed 95% confidence interval. The heterogeneity indicators (I^2^, τ^2^, *p*) are presented below each random effects Forest plots.

**Figure 2 medicina-58-00060-f002:**
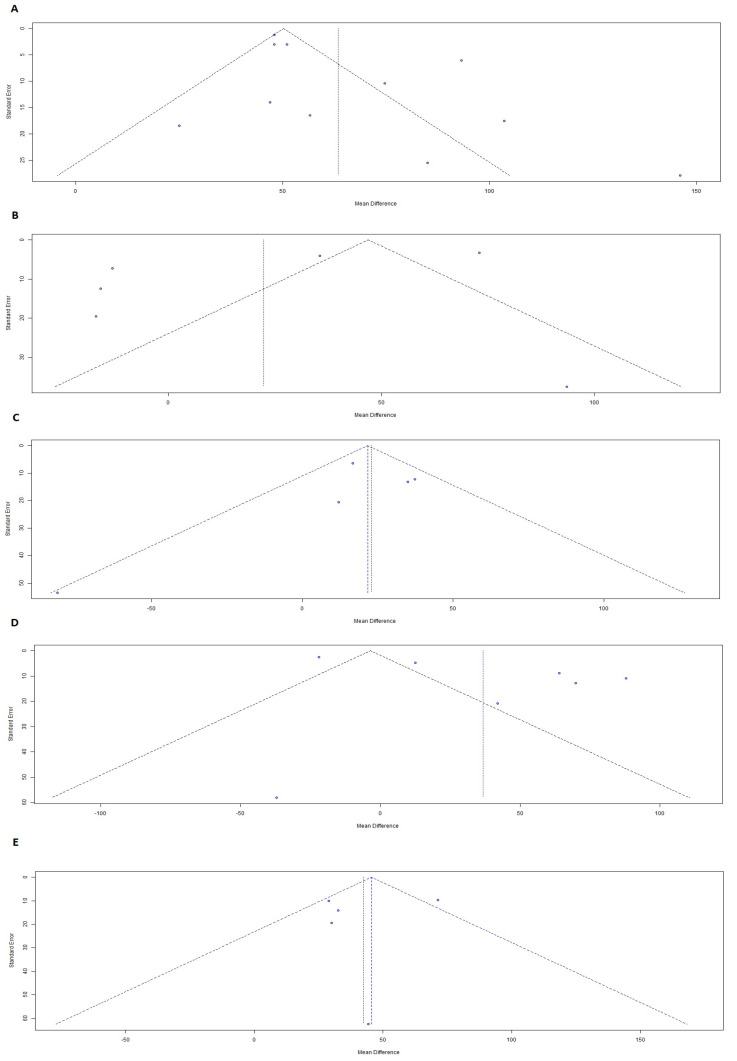
Funnel plots for YKL-40 CSF levels in AD versus NC (**A**), MCI versus NC (**B**), MCIAD versus MCI (**C**), AD versus MCI (**D**), and AD versus MCIAD (**E**) indicating no publication bias according to Egger’s linear regression test. The circles indicate the individual studies included in this analysis. The black dotted line indicated the combined standard error of the mean differences, while the blue dotted line indicated the 95% confidence intervals.

**Figure 3 medicina-58-00060-f003:**
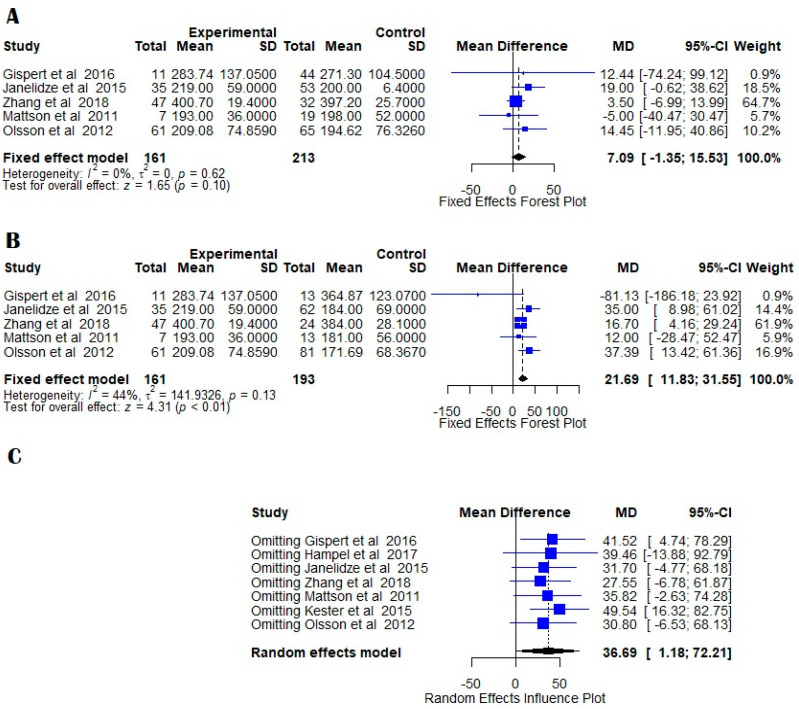
Forest plots of YKL-40 CSF levels in MCIAD versus NC; (**A**) and MCIAD versus MCI; (**B**) indicating decreased heterogeneity in fixed random model. The figures indicate the means, standard errors (pg/mL), 95% confidence intervals, and weighted mean differences for each study included in this meta-analysis. For each comparison, a supplemental random effects influence plot was designed to further evaluate the possible publication bias when omitting each of the included studies; (**C**) mean differences and 95% confidence intervals being calculated in each situation. The standardized mean differences between patients and controls are represented as squares whose sizes are proportional to the sample sizes. The whiskers represent the 95% confidence intervals. The diamonds represent the pooled estimate based on the random-effect model with the center representing the point estimate and the width indicating the assessed 95% confidence interval. The heterogeneity indicators (I^2^, τ^2^, *p*) are presented below each random effects Forest plots.

**Figure 4 medicina-58-00060-f004:**
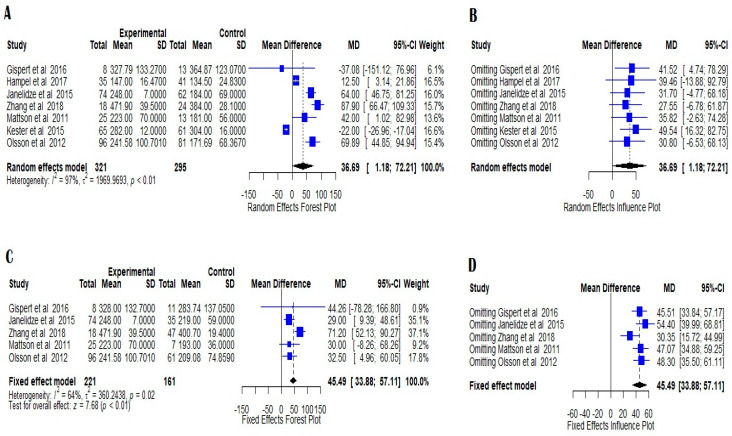
Forest plots of YKL-40 CSF levels in AD versus MCI; (**A**) and AD versus MCIAD; (**C**) indicating high heterogeneity in random (AD versus MCI) and fixed effects models (AD versus MCIAD). The figures indicate the means, standard errors (pg/mL), 95% confidence intervals, and weighted mean differences for each study included in this meta-analysis. For each comparison, a supplemental random effects influence plot was designed to further evaluate the possible publication bias when omitting each of the included studies; (**B**,**D**) mean differences and 95% confidence intervals being calculated in each situation. The standardized mean differences between patients and controls are represented as squares whose sizes are proportional to the sample sizes. The whiskers represent the 95% confidence intervals. The diamonds represent the pooled estimate based on the random-effect model with the center representing the point estimate and the width indicating the assessed 95% confidence interval. The heterogeneity indicators (I^2^, τ^2^, *p*) are presented below each random effects Forest plots.

## Data Availability

The datasets used and analyzed during the current study are available from the corresponding author on reasonable request.

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
