# Peer review of "YKL-40 as a Potential Biomarker for the Differential Diagnosis of Alzheimer’s Disease"

_medicina, 2021, doi:10.3390/medicina58010060_

Round 1

Reviewer 1 Report

The authors in the present study investigated well the role of the YKL-40 levels in the CSF for the diagnosis of the AD patients and also they have discussed whether there are further potential role for this protein for managing and treatment of the AD.

1.The research in this paper is relevant and very interesting.
2.This topic that the authors have chosen is original.
3.The paper as I have indicated in my review is well written and easy to read.
4.Their conclusions are well consistent with their arguments presented in theit text.

Author Response

Reviewer  1
The authors in the present study investigated well the role of the YKL-40 levels in the CSF for the diagnosis of the AD patients and also they have discussed whether there are further potential role for this protein for managing and treatment of the AD.

1.The research in this paper is relevant and very interesting.
2.This topic that the authors have chosen is original.
3.The paper as I have indicated in my review is well written and easy to read.
4.Their conclusions are well consistent with their arguments presented in their text.

Thank you for your kind suggestions and review!

Reviewer 2 Report

Medicina; medicina-1478537

Title: YKL-40 as a potential biomarker for the differential diagnosis of Alzheimer’s disease

Mavroudis and co-authors conducted a literature study to determine whether cerebrospinal fluid (CSF) levels of YKL-40 serve as a reliable indicator of the progression of Alzheimer’s disease (AD). In brief, the authors surveyed peer-reviewed manuscripts published from 1990 to 2021 using the PubMed, Web of Science, and Cochrane databases. As a result of their research, the authors conclude that YKL-40 could be potentially useful as a biomarker to delineate mild cognitive impairment and AD. See comments below.

(1) The primary concern is that there are already numerous other studies (e.g., PMIDs 28183245, 27068280, 28281838, 31794792, 2568193) that explore YKL-40 as a biomarker for tracking AD prognosis. As presented, it is not clear how the current study builds from prior research to provide new information.

(2) The Figures are not of sufficient quality for viewing and interpretation.

(3) In the Discussion, reasons for similarities or differences among individual studies (e.g., Mattson et al.) have not been adequately resolved concerning YKL-40 as reliable biomarker for MCI and/or AD.

(4) All abbreviations must be defined at first mention in the primary body of the manuscript.

Author Response

Reviewer  2

Title: YKL-40 as a potential biomarker for the differential diagnosis of Alzheimer’s disease

Mavroudis and co-authors conducted a literature study to determine whether cerebrospinal fluid (CSF) levels of YKL-40 serve as a reliable indicator of the progression of Alzheimer’s disease (AD). In brief, the authors surveyed peer-reviewed manuscripts published from 1990 to 2021 using the PubMed, Web of Science, and Cochrane databases. As a result of their research, the authors conclude that YKL-40 could be potentially useful as a biomarker to delineate mild cognitive impairment and AD. See comments below.

(1) The primary concern is that there are already numerous other studies (e.g., PMIDs 28183245, 27068280, 28281838, 31794792, 2568193) that explore YKL-40 as a biomarker for tracking AD prognosis. As presented, it is not clear how the current study builds from prior research to provide new information.

Besides confirming the previous described relevance of YKL-40 as a relevant marker in AD, we do believe that the present study do provides new insights on the discussed matter, as more studies are included in the analysis, as compared to the studies kindly indicated by the reviewer (we do added now in the Methods sections that studies are updated to October 2021).

(2) The Figures are not of sufficient quality for viewing and interpretation.

 Figures were updated graphically, focusing especially on figure 2.

(3) In the Discussion, reasons for similarities or differences among individual studies (e.g., Mattson et al.) have not been adequately resolved concerning YKL-40 as reliable biomarker for MCI and/or AD.

We added that “Regarding the controversies and the reasons for the differences among individual studies, we can add that, for example Mattsson et al. (2017) described increased levels of YKL-40 in the AD patients, when compared to controls and patients with stable MCI, but they did not find a statistically significant difference between groups (as mentioned above), while also different markers analyzed by this research group did not correlate with each other, except for YKL-40 versus CCL2 (chemokine C-C motif ligand 2) in patients with other dementias [21]. Also, there were no modifications of YKL-40 in AD or in prodromal AD, while in the Craig-Schapiro study from 2010 [25] the cohort was larger than in Mattson article, with the reported group differences “being small with large overlaps, making the findings difficult to reproduce”, as stated by [21].”

(4) All abbreviations must be defined at first mention in the primary body of the manuscript.

An abbreviation list was added. Thank you

Reviewer 3 Report

29th November, 2021

Review of the Manuscript ID: medicina-1478537, by I. Mavroudis et al., entitled: “YKL-40 as a potential biomarker for the differential diagnosis of Alzheimer’s disease” that is intended to be published as the Review in Medicina

(separate Microsoft Word file as Reviewer Attachment for Manuscript ID medicina-1478537 Medicina 29th November 2021 that includes Comments to the Authors is also uploaded)

Taking into consideration research highlight, contribution of the Authors to the progress in the research field, thorough manner of data presentation, very well writing in English, abundance of Results and Figures (diligent graphic visualization), the quality of this paper deserves praise and merits my support. The Authors have received the high scores from me for the originality, importance of the work and the scientific value of their paper. In my opinion, the current paper provides insightful interpretation of topical and coming trends in precisely identifying cerebrospinal fluid (CSF)-specific and highly predictable/prognostic biomarkers related to YKL-40 chitin-binding lectin-dependent and differential diagnostic approaches that can be used to properly recognize and distinguish the multifaceted etiopathogenesis of such severe (chronic and acute) neurodegenerative disorders as Alzheimer’s disease (AD). This could enable neurologists to develop and optimize the novel anti-neuroinflammatory therapeutic strategies and potent personalized treatments in patients afflicted with AD. For all the above-indicated reasons, I strongly recommend the Editorial Board to allow for publication of this valuable Review in Medicina, after the minor revision of the manuscript will have been completed by the Authors and provided that the Authors are ready to consider all the Reviewer comments shown below:

1) There is a lack of the separate Abbreviations section in the paper. That is why, this section should have been added at the end of the manuscript to comprehensively elucidate and expand a wide range of the in-text abbreviations, which have been used by the Authors in all the subsections of their paper.

2) The References section has to be prepared in the format compatible with the requirements of Medicina.

General Comment of the Reviewer:

Before the manuscript will have been accepted for publication in Medicina, it requires the minor revision (according to all the aforementioned recommendations and suggestions of the Reviewer).

Author Response

Reviewer 3
Review of the Manuscript ID: medicina-1478537, by I. Mavroudis et al., entitled: “YKL-40 as a potential biomarker for the differential diagnosis of Alzheimer’s disease” that is intended to be published as the Review in Medicina

Taking into consideration research highlight, contribution of the Authors to the progress in the research field, thorough manner of data presentation, very well writing in English, abundance of Results and Figures (diligent graphic visualization), the quality of this paper deserves praise and merits my support. The Authors have received the high scores from me for the originality, importance of the work and the scientific value of their paper. In my opinion, the current paper provides insightful interpretation of topical and coming trends in precisely identifying cerebrospinal fluid (CSF)-specific and highly predictable/prognostic biomarkers related to YKL-40 chitin-binding lectin-dependent and differential diagnostic approaches that can be used to properly recognize and distinguish the multifaceted etiopathogenesis of such severe (chronic and acute) neurodegenerative disorders as Alzheimer’s disease (AD). This could enable neurologists to develop and optimize the novel anti-neuroinflammatory therapeutic strategies and potent personalized treatments in patients afflicted with AD. For all the above-indicated reasons, I strongly recommend the Editorial Board to allow for publication of this valuable Review in Medicina, after the minor revision of the manuscript will have been completed by the Authors and provided that the Authors are ready to consider all the Reviewer comments shown below:

1) There is a lack of the separate Abbreviations section in the paper. That is why, this section should have been added at the end of the manuscript to comprehensively elucidate and expand a wide range of the in-text abbreviations, which have been used by the Authors in all the subsections of their paper.

An abbreviation list was added. Thank you.

2) The References section has to be prepared in the format compatible with the requirements of Medicina.

References were done according to the requirements of the Medicina. Please accept our apologies for missing that initially.

Round 2

Reviewer 2 Report

The authors modified the manuscript but major concerns remain. It would help to have clear study objectives in the Introduction while indicating precisely how they aimed to build from and/or complement other published studies. What exactly are the new insights provided by this study versus prior publications? A mere literature search update to 10/2021 doesn't seem sufficient. Also, the legibility of Figures has improved but Figure Legends lack informative value and statistical terms (e.g., I2, tau2, z-value) have not been defined well in the context of the author's interpretation/conclusions.   

Author Response

24th December 2021

Dear Assistant Editor, Ms. Arya Huang,

We are writing regarding the medicina-1478537 manuscript, entitled ”YKL-40 as a potential biomarker for the differential diagnosis of Alzheimer’s disease”, which received a major revision on the 17th December 2021. We are hereby responding to the kind suggestions of the reviewer: ”The authors modified the manuscript but major concerns remain. It would help to have clear study objectives in the Introduction while indicating precisely how they aimed to build from and/or complement other published studies. What exactly are the new insights provided by this study versus prior publications? A mere literature search update to 10/2021 doesn't seem sufficient. Also, the legibility of Figures has improved but Figure Legends lack informative value and statistical terms (e.g., I2, tau2, z-value) have not been defined well in the context of the author's interpretation/conclusions.”

Thank you for your kind suggestions which helped us to improve our work. According to your perspective, we decided that the last paragraph from the Introduction needed improvement, thus we added a more precise aim of the study and discussed the way we planned to complement other published studies. In this way, considering that the reviewer previously indicated some papers by their PMID in which the main subject was the YKL-40 possible importance as a neurodegeneration marker (meta-analyses and original studies), we commented on the novelty of our study which resides in the fact that, in our best knowledge, no previous meta-analysis evaluated YKL-40 importance in the context of disease progression (from normal to stable MCI, to progressive MCI, to AD): ”Previous reports regarding the possible use of YKL-40 CSF levels in the diagnosis of AD were focusing on the comparison between healthy individuals versus AD or other neurodegenerative diseases patients [14-16]. However, a recent study regarding the CSF levels of YKL-40 in prodromal AD [17] demonstrated that YKL-40 CSF levels show differences in AD versus MCI patients implying that this inflammatory marker could predict the conversion of MCI in AD, especially in the presence of APOE ε4 allele. Thus, based on these aspects, we aim to analyse the YKL-40 CSF levels power to predict neurodegeneration (specific to MCI, AD, or MCI-AD), and to evaluate its potential for differential diagnosis and therapeutical response.”

Also, we decided that the figure legends needed further improvement, thus we added some informative value to the legends and we described the statistical parameters and the used symbols. For example, ”Figure 1. Forest plots of YKL-40 CSF levels in AD versus NC (A) and MCI versus NC (C) indicating high heterogeneity in random effect model and statistically significant differences between the compared groups. The figures indicate the means, standard errors (pg/mL), 95% confidence inter-vals, and weighted mean differences for each study included in this meta-analysis. For each com-parison, a supplemental random effects influence plot was designed to further evaluate the possible publication bias when omitting each of the included studies (B, D), mean differences and 95% confidence intervals being calculated in each situation. No statistically significant impact was ob-served when omitting each of the evaluated studies. The standardized mean differences between patients and controls are represented as squares whose sized are proportional to the sample sizes. The whiskers represent the 95% confidence intervals. The diamonds represent the pooled estimate based on the random-effect model with the center representing the point estimate and the width indicating the assessed 95% confidence interval. The heterogeneity indicators (I2, τ2, p) are presented below each random effects Forest plots.”

Further improvement was brought by some text corrections (typos and grammar), refining the references section, and abbreviations list update.

Kind regards,

Alin.

This manuscript is a resubmission of an earlier submission. The following is a list of the peer review reports and author responses from that submission.